# OpenReview forum: "Response Tuning: Aligning Large Language Models without Instruction"
_ICLR.cc/2025/Conference — Submitted to ICLR 2025_

### Official Review · Reviewer_krvS · 2024-10-30

**Soundness:** 3
**Presentation:** 3
**Contribution:** 3
**Rating:** 5
**Confidence:** 4

**Summary:**

Based on the hypothesis that language models learn to follow instructions during instruction tuning mainly by learning the output space, the paper presents experiments where models are trained only on responses ("Response Tuning" or RT) instead of conditioning them on instructions (the usual "Instruction Tuning" or IT).

RT on various Llama and Gemma models with data from Alpaca, Dolly, and LIMA performs comparable to IT on instruction following (AlpacaEval) and several core capability evaluations (MMLU, OpenbookQA, Hellaswag, ARC, GSM8K, and PIQA).

To further demonstrate the importance of the quality of the responses, the responses from the datasets above are refined using a larger LM, and training on this refined dataset improves AlpacaEval scores in both IT and RT setups.

When trained on LM responses corresponding to safety related refusals, though IT performs better than RT initially, with increasing amounts of data, performance of RT models eventually matches IT.

In an ICL (instead of finetuning) setup as well, it is shown that LMs can be aligned with human preferences using response demonstrations alone, with response-only ICL yielding comparable performance to paired-demonstration ICL in one evaluation setting (https://allenai.github.io/re-align/just_eval.html).

**Strengths:**

The main finding in this paper is interesting. It provides an insight into how LMs learn to follow instructions. Future work may be able to leverage this insight to improve instruction tuning procedures.

**Weaknesses:**

## Practical utility
The paper does not demonstrate the practical utility of the finding. It would be helpful in the following cases:

1. Obtaining instruction-response pairs is significantly harder than obtaining responses alone.
2. Training on responses alone is more effective than training on instruction-response pairs.

1 is not true in any of the experimental settings considered in the paper, and 2 is not true in any of the results, at least not significantly.

Suggestions for addressing this concern:

- There may be tasks where 1 is true. Code generation (from natural language instructions) is a good example. Does RT on code, which is much easier to obtain than paired NL-code examples do as well as IT? More generally, tasks with structured outputs that are easy to synthetically generate are suitable here.
- Tasks where the model performance is sensitive to the superficial features in the instructions would be helpful to demonstrate 2. Does RT make the model more robust to variations in instructions than IT in such cases?

## Role of the response refinement experiments
These experiments generally show the importance of the quality of the responses, but it is unclear how this is related to the main hypothesis about conditioning on instructions being unnecessary. Refinement generally helps in both IT and RT and RT settings. It would be more helpful to directly compare IT and RT models with and without response refinement. Does refinement increase the winrate of RT models against IT models?

**Questions:**

- What is the performance of the base models (Llama 3.1 8B and Gemma 2 9B) on the tasks shown in Table 1?
- (More of a suggestion) It would be easier to interpret the results in Table 3 if all the models (IT and RT with and without response refinement) are evaluated against a fixed reference model for AlpacaEval.

---

> ### Author Response · Authors · 2024-11-22
> **Response to Reviewer krvS**
>
> We sincerely appreciate your insightful and constructive feedback. Below, we have addressed each of your comments and implemented the suggested revisions in the manuscript.
>
> ---
>
> **1. Practical Utility**
>
> We appreciate your constructive feedback and insightful suggestions for demonstrating the practicality of RT. In response, we conducted additional experiments using code data and presented the results in our general response 2, along with further discussion on the practicality of RT based on guidance from other reviewers.
>
> To compare the robustness to instruction variations between IT and RT models, we performed a pairwise evaluation on models trained on the code mentioned above using the AlpacaEval test set, which consists of general queries. The results below indicate that RT models may not be more robust than IT models.
>
> |  | AlpacaEval win-rate against Code IT model |
> | --- | --- |
> | Code RT | 42% |
>
> While these practical considerations are valuable, we kindly reiterate that the primary value of our study lies in the insights derived from the experimental results, which aim to deepen the understanding of how language models acquire capabilities needed to handle instructions.
>
> **2. (and 4) Role of the response refinement experiments**
>
> Thank you for your question regarding the response refinement experiment. The experiments in Section 5 were designed to investigate whether adjusting the response distribution can help achieve desired alignment objectives, such as improved preference alignment. To ensure a controlled comparison, we evaluated IT and RT models independently before and after refinement, allowing for clearer comparisons within each setup.
>
> As per your suggestion, we additionally compared IT (refined) vs. RT (refined) models and found trends similar to those observed in the initial IT (original) vs. RT (original) comparisons in Section 4. Notably, when comparing the IT (original) model with the RT (refined) model, we observed that the RT model was preferred, emphasizing the critical role of response quality in alignment. The following table shows the AlpacaEval win rates for the Llama-3.1-8B RT model trained using refined LIMA responses against IT models:
>
> |  | vs IT (using refined responses) | vs IT (using original responses) |
> | --- | --- | --- |
> | RT using refined responses | 40% | 53% |
>
> **3. What is the performance of the base models (Llama 3.1 8B and Gemma 2 9B) on the tasks shown in Table 1?**
>
> Thank you for the question. We have included evaluation results for the untuned base models in Tables 1 and 11 to provide a clearer comparison point. The specific results for the Llama and Gemma models are as follows:
>
> |  | MMLU | OpenbookQA | HellaSwag | ARC | GSM8K | PIQA | Avg. |
> | --- | --- | --- | --- | --- | --- | --- | --- |
> | Llama-3.1-8B Base | 63.36 | 33.6 | 60.04 | 66.34 | 55.72 | 80.14 | 59.87 |
> | Gemma-2-9B Base | 69.04 | 33.8 | 61.09 | 74.42 | 69.9 | 81.28 | 64.92 |
>
> Additionally, we have expanded our explanation of these results in the paper. Specifically, we observe that both IT and RT models demonstrate a similar and slight level of performance drop [1], while largely retaining the core capabilities of their pre-trained base models.
>
> ---
>
> **References**
>
> [1] Fu et al. "Disperse-Then-Merge: Pushing the Limits of Instruction Tuning via Alignment Tax Reduction." Findings of ACL 2024.

---

> > ### Comment · Reviewer_krvS · 2024-11-25
> >
> > I thank the authors for their detailed response and for presenting results from additional experiments.
> >
> > ## Results from base models in Tables 1 and 11
> > These results are informative, but raise questions about the core claims in the paper. The trend is that in almost all the evaluations considered, the base models seem to perform better than IT and RT. I interpret that the models are not actually learning the tasks during finetuning, either from IT or RT. Is this a setting where one can make meaningful comparisons between finetuning methods?
> >
> > ## HumanEvalPack results
> > It is nice to see that RT and IT perform comparably in this setting, but to interpret these results, I think one would need to compare against a baseline, e.g., the base model's performance like above, or IT on a portion of the training data. Essentially the main question is how much of the performance *gain* we see from IT are we recovering from RT?
> >
> > ## Response refinement
> > From the new results presented in the response, I see that response refinement does not reduce the gap between IT and RT, but actually increases it. Is it right to say that in the context of the main claims in the paper (RT yields models comparable to IT), response refinement is not relevant?
> >
> > I would like to hear from the authors to confirm that I am not misunderstanding or misinterpreting the new results before I adjust my score.

---

> > > ### Author Response · Authors · 2024-11-26
> > > **Follow-Up Response to Reviewer krvS (1/2)**
> > >
> > > Thank you for your follow-up feedback! We’re happy to provide further explanation on the additional results.
> > >
> > > **1. Results from the Base Models**
> > >
> > > Thank you for your question. We would like to kindly remind that the purpose of IT (or RT) is to make models to generate responses following instructions. While IT/RT does not primarily aim at knowledge acquisition, it is important to validate that the resulting instruction-following models retain the knowledge foundation required to perform instructed tasks. In light of this, the core capabilities evaluation setup was proposed in previous work [1], and we have conducted this evaluation to assess RT models in this regard. To clarify, we have enhanced the explanation of the evaluation's background in the updated manuscript (line 267).
> > >
> > > As a result, we found that the performance of RT models is comparable to IT models, indicating that RT models retain similar core capabilities to IT models, while both exhibit slight degradation from the base models—a phenomenon known as alignment tax, where tuned models may lose some pre-training knowledge due to parameter updates during fine-tuning [1, 2].
> > >
> > > **2. Regarding HumanEvalPack results**
> > >
> > > We appreciate your question and thoughtful suggestion. HumanEvalPack is specifically designed to evaluate *instruction-following models*. Its evaluation queries are framed as *instructions* (as cited directly from their work: *This work studies training and evaluation of Code LLMs that follow instructions.* [3]). Here is an example of an evaluation query:
> > >
> > > ```latex
> > > Write a Python function `has_close_elements(numbers: List[float], threshold: float) -> bool` to solve the following problem:
> > > Check if in given list of numbers, are any two numbers closer to each other than
> > > given threshold.
> > > >>> has_close_elements([1.0, 2.0, 3.0], 0.5)
> > > False
> > > >>> has_close_elements([1.0, 2.8, 3.0, 4.0, 5.0, 2.0], 0.3)
> > > True
> > > ```
> > >
> > > Since base models lack the *capability to generate responses* to instructions—including code-generation instructions—using HumanEvalPack to evaluate untuned models may not be fair or meaningful. Additionally, we confirm that base models score **zero** under the same evaluation setup, consistent with our findings from instructability evaluations. Here is an example of the base model output for a code-generation instruction, which produces meaningless output tokens:
> > >
> > > ```latex
> > > <|problem|>\n<|solution|>\n<|feedback|>\n<|end|>
> > > ```
> > >
> > > Therefore, we believe the current evaluation setup, which **uses the IT model trained on the same code data but paired with instructions as the baseline,** is appropriate for demonstrating the code-generation instruction-following capabilities of RT models.
> > >
> > > **3. Regarding response refinement experiment**
> > >
> > > We would like to clarify the results of the experiment. Below are the win rates for the Llama-3.1-8B RT model trained using **original** LIMA responses (Figure 4b, line 275):
> > >
> > > |  | vs IT model (trained using **original** responses) |
> > > | --- | --- |
> > > | Win-rate of RT model (trained using **original** responses) | 40% |
> > >
> > > Below are the win rates for the Llama-3.1-8B RT model trained using **refined** LIMA responses:
> > >
> > > |  | vs IT model (trained using **refined** responses) | vs IT model (trained using **original** responses) |
> > > | --- | --- | --- |
> > > | Win-rate of RT model (trained using **refined** responses) | 40% **(remains the same)** | 53% |
> > >
> > > As shown above, response refinement **for both models** does not reduce the performance gap between IT and RT models but instead **remains unchanged**. This is expected, as both methods use identical training responses. However, when comparing the **RT model trained with refined responses** to the **IT model trained with original responses,** the RT model outperforms the IT model due to the refinement of its training responses.
> > >
> > > The comparable performance of RT demonstrated in Section 4 (“RT yields models comparable to IT”) implies that establishing a response distribution is important for achieving instruction-following alignment. Building on this, the refinement experiment in Section 5 further explores and corroborates the significance of the response distribution. Thus, we view this experiment as a derived and extended study of Section 4.
> > >
> > > However, we acknowledge that this experiment is somewhat indirect in relation to the main claims, which focus more on the “importance of paired instructions in gaining specific functionality (instruction-following and safety).” As a result, we are considering rearranging this section by swapping its order with the In-Context Response Learning section (Section 7).
> > >
> > > We would appreciate any feedback you may have and would be happy to incorporate it to further enhance our work.

---

> > > ### Author Response · Authors · 2024-11-26
> > > **Follow-Up Response to Reviewer krvS (2/2)**
> > >
> > > **References**
> > >
> > > [1] Wang et al. "How far can camels go? exploring the state of instruction tuning on open resources." NeurIPS 2023.
> > >
> > > [2] Fu et al. "Disperse-Then-Merge: Pushing the Limits of Instruction Tuning via Alignment Tax Reduction." Findings of ACL 2024.
> > >
> > > [3] Muennighoff et al. "Octopack: Instruction tuning code large language models." ICLR 2024.

---

> > > ### Author Response · Authors · 2024-11-28
> > > **Looking Forward to Further Discussions**
> > >
> > > Dear Reviewer krvS,
> > >
> > > We sincerely appreciate the time and effort you have devoted to reviewing our paper and providing valuable feedback. We have carefully considered your comments and endeavored to address your concerns and questions. We kindly ask if our responses and the updated manuscript adequately address them. If so, we would like to respectfully request your reconsideration of the rating.
> > >
> > > If there are any areas requiring further clarification or additional explanation, please do not hesitate to let us know. We would be happy to address any additional comments you may have. Thank you once again for your thoughtful review and efforts.

---

> ### Comment · Reviewer_krvS · 2024-11-29
>
> I thank the authors for further clarifying the new results. Based on my current understanding, I believe the paper needs some reorganization and modification in experimental settings as elaborated below. Hence I am leaving my score unchanged.
>
> ## Relevant empirical settings
> As I wrote earlier, I think the main question that should be answered by the experiments is how much of the boost in model performance resulting from IT can RT also provide. So settings where IT does not provide a boost (i.e., where the performance of IT model is worse that of the base model) are not relevant, and I think the core capability results in Tables 1 and 11 are not adding much to the paper.
>
> Also, I do not think IT models' performance being worse than the base models' is a fundamental issue ("alignment tax") of the setup. It may simply be that the instruction tuning datasets chosen for the experiments are not helpful for the evaluation tasks. For example, it is reported that the Tulu 2 instruction tuning mix leads to improvements over the Llama 3.1 8B base model in several core evaluations (see https://huggingface.co/allenai/llama-3.1-tulu-2-8b). I would recommend choosing an instruction tuning mix and relevant evaluations where IT improves performance over the base model (e.g. the Tulu 2 setup) to compare IT vs. RT.
>
> The new coding results are actually relevant in this setup. I agree with the authors that the coding task is not designed for non-instruction tuned models, but in the context of this paper, I think it is fair to compare base models with IT vs. RT. It is quite interesting that RT yields performance close to IT in this setting. I think this result can be analyzed further.
>
> ## Response refinement experiments
> I thank the authors for clarifying this result. I now see that response refinement does not change the difference between IT and RT, and I believe we are in agreement that this result is not directly related to the core thesis of the paper.

---

> ### Author Response · Authors · 2024-11-30
> **Follow-Up Response to Reviewer krvS**
>
> Thank you for your engagement and thoughtful comments. We have addressed each of your comments in detail below.
>
> **1.1. On the core capabilities evaluation**
>
> Thank you for raising this concern. We would like to clarify the scope of our study and the goal of the core capabilities evaluation within it.
>
> The primary focus of our study lies in the ability to handle instructions achieved through IT/RT, as indicated by the paper's title and our explanation within the paper. Therefore, we believe that investigating specific downstream task performance using particular datasets (e.g., the full Tulu mixture you referenced or a CoT subset aimed at improving mathematical reasoning [1]) falls beyond the scope of our study. This might represent a promising future direction.
>
> We conducted the core capabilities evaluation as a supplementary assessment to the instructability evaluation, aiming to verify the validity of the instructability gained through RT (line 267). If RT results in significant knowledge loss—leading to an instructable but knowledge-lacking assistant—the trained model would be invalid. This evaluation seeks to address this concern, and the results show that RT models perform comparably to IT baselines in this regard. We believe this evaluation is essential to fully demonstrate that RT models are both instructable and knowledgeable assistants.
>
> **1.2. On the code-generation instruction-following evaluation**
>
> Thank you once again for suggesting this thoughtful experiment. As you can see, this is an interesting experimental result that further supports our hypothesis. We included it in the previous revision to discuss the practicality of RT, but we plan to provide additional explanation to clarify its relevance to our hypothesis. Additionally, we would like to remind you that we have already performed this evaluation for the base model, which scored zero. Since the base model is not instructable, we did not observe any valid or correct code from it without explicitly forcing the model to generate code, as attempted by Reviewer AfZU.
>
> |  | HumanEvalPack (Pass@10) |
> | --- | --- |
> | Llama-3.1-8B IT (using code snippets) | 0.45 |
> | Llama-3.1-8B RT (using code snippets) | 0.40 |
> | Llama-3.1-8B Base | 0 |
>
> **2. On the response refinement experiments**
>
> We appreciate your guidance. While we believe this experiment provides extended and relevant insights within our study, adjusting its position would allow for a more focused discussion on our main hypothesis. As mentioned earlier, and in line with your suggestion, we will swap the order of Sections 5 and 7 to improve the presentation. Given that the paper revision period has ended, we will incorporate these updates into the final version.
>
> ---
>
> **References**
>
> [1] Wang et al. "How far can camels go? exploring the state of instruction tuning on open resources." NeurIPS 2023.

---

> > ### Author Response · Authors · 2024-12-02
> > **Awaiting Your Valuable Feedback**
> >
> > Dear Reviewer krvS,
> >
> > We deeply appreciate the time and effort you’ve dedicated to the review process. This is a gentle reminder that there are 24 hours remaining for you to provide any additional comments or feedback on our paper.
> >
> > We look forward to receiving your further comments. If there are any remaining questions or areas requiring clarification, please don’t hesitate to share them—we would be happy to provide any additional information you may need. If our previous responses have sufficiently addressed your concerns, we kindly request you to consider updating your evaluation.
> >
> > Thank you once again for your thoughtful contributions.

---

> > > ### Comment · Reviewer_krvS · 2024-12-02
> > >
> > > I thank the authors for the engagement so far. I have no further questions.
> > >
> > > Regarding my suggestion of using other instruction tuning mixes, it was not about investigating downstream performance in specific tasks, but about running experiments in a setting that is practically relevant (i.e., IT actually helps), and I believe that is within the scope of this study.
> > >
> > > Since I believe these experiments need to be updated in the paper, and also given the other structural changes we discussed so far, I am keeping my current score.

---

> ### Author Response · Authors · 2024-12-03
> **Appreciation and Final Remarks**
>
> We truly appreciate your continuous engagement and thoughtful feedback! In light of your final comments, we would like to share our thoughts and emphasize our willingness to improve our work based on your remaining suggestions.
>
> For the core capabilities evaluation, we would like to reiterate that its goal is to validate whether RT models retain a similar amount of knowledge as IT models, which serves as the foundational knowledge required to perform instructed tasks. In light of this purpose, we believe that a comparison between IT and RT remains meaningful, regardless of whether IT models outperform or underperform the base model. Therefore, we consider our current setup valid.
>
> Regarding the discussed structural change, we are prepared to incorporate relevant modifications in our camera-ready version. We believe the refinement experiment offers meaningful insights into the response distribution, and we will further optimize the presentation of these insights based on your suggestions.
>
> Thank you once again for the invaluable time you have dedicated throughout the review process.

---

### Official Review · Reviewer_JXF8 · 2024-11-03

**Soundness:** 3
**Presentation:** 3
**Contribution:** 3
**Rating:** 5
**Confidence:** 4

**Summary:**

The article suggests that the model's instruction-following capability can be acquired by fine-tuning using with the response data, and this hypothesis is validated through experiments. Furthermore, the article demonstrates enhanced preference alignment and safety by controlling the response distribution.

**Strengths:**

1. The article is well-written and highly readable.
2. It presents an interesting hypothesis and proposes a simple yet effective method.
3. Experiments were conducted across various benchmarks.

**Weaknesses:**

1. The pre-training data for some models in the experiments not publicly available, and models like Gemma-2 already possess some instruction-following capability. This suggests that the pre-training data may inherently include instruction-response pairs, potentially skewing the experimental results and affecting accuracy.
2. Experimental results show that the performance of Response Tuning still falls short of Instruction Tuning, particularly in human and GPT-4 evaluations. Although this is an interesting finding, deliberately constructing a response set without instructions for training seems neither effective nor efficient. Therefore, the significance of this method for advancing research remains questionable.

**Questions:**

1. Can you conduct similar experiments using models with publicly available pre-training data, such as OLMo, TinyLlama, and Pythia, to validate the conclusion of your article?
2. In Table 1, why does Instruction Tuning perform better on several benchmarks but have a lower overall performance compared to Response Tuning? How is the overall performance in Table 1 calculated? For instance, the arithmetic averages for Response Tuning on these benchmarks are 54.05 and 61.08, so why are they reported as 58.42 and 65.62?

---

> ### Author Response · Authors · 2024-11-22
> **Response to Reviewer JXF8**
>
> We sincerely appreciate your thoughtful feedback and insightful suggestions. Below, we address each of your concerns and highlight the corresponding updates incorporated into the revised manuscript.
>
> ---
>
> **Response to weaknesses:**
>
> **1. The pre-training data for some models in the experiments not publicly available, and models like Gemma-2 already possess some instruction-following capability. This suggests that the pre-training data may inherently include instruction-response pairs, potentially skewing the experimental results and affecting accuracy.**
>
> Thank you for raising this important concern. We carefully considered this issue during the experimental design phase. To address it, we included human evaluation results for all untuned base models in Section 4. These results (Figure 2 and Table 10) demonstrate that none of the four models function as instruction-following models without establishing a response distribution. In the revised manuscript, we have added further explanation on these experimental results, as suggested by Reviewer 6b1i, to provide additional clarity and context (line 303).
>
> **2. Experimental results show that the performance of Response Tuning still falls short of Instruction Tuning, particularly in human and GPT-4 evaluations. Although this is an interesting finding, deliberately constructing a response set without instructions for training seems neither effective nor efficient. Therefore, the significance of this method for advancing research remains questionable.**
>
> Thank you for raising this point. In general response 1 and 3, we have provided additional clarification on the scope and objectives of our work, as well as further analysis of the pairwise evaluation results you highlighted. We hope these additions address your concerns.
>
> **Response to questions:**
>
> **1. Can you conduct similar experiments using models with publicly available pre-training data, such as OLMo, TinyLlama, and Pythia, to validate the conclusion of your article?**
>
> Thank you for the constructive suggestion. In response, we conducted additional experiments using OLMo and included the results in the general response 4. The results indicate that the main trends observed in our study are consistent with OLMo.
>
> **2. How is the overall performance in Table 1 calculated? For instance, the arithmetic averages for Response Tuning on these benchmarks are 54.05 and 61.08, so why are they reported as 58.42 and 65.62?**
>
> Thank you for identifying this discrepancy. We apologize for the oversight. The "Overall" column in Table 1 is intended to represent the arithmetic mean of all benchmark scores. Upon review, we found that OpenBookQA scores were inadvertently omitted from the calculations for all rows except the first one. We have recalculated the overall scores, corrected the affected tables, and updated the manuscript accordingly to ensure accuracy (Tables 1 and 11).

---

> ### Author Response · Authors · 2024-11-28
> **Looking Forward to Further Discussions**
>
> Dear Reviewer JXF8,
>
> We sincerely appreciate the time and effort you have devoted to reviewing our paper and providing valuable feedback. We have carefully considered your comments and endeavored to address your concerns and questions. We kindly ask if our responses and the updated manuscript adequately address them. If so, we would like to respectfully request your reconsideration of the rating.
>
> If there are any areas requiring further clarification or additional explanation, please do not hesitate to let us know. We would be happy to address any additional comments you may have. Thank you once again for your thoughtful review and efforts.

---

> > ### Author Response · Authors · 2024-12-02
> > **Awaiting Your Valuable Feedback**
> >
> > Dear Reviewer JXF8,
> >
> > We deeply appreciate the time and effort you’ve dedicated to the review process. This is a gentle reminder that there are 24 hours remaining for you to provide any additional comments or feedback on our paper.
> >
> > We look forward to receiving your further comments. If there are any remaining questions or areas requiring clarification, please don’t hesitate to share them—we would be happy to provide any additional information you may need. If our previous responses have sufficiently addressed your concerns, we kindly request you to consider updating your evaluation.
> >
> > Thank you once again for your thoughtful contributions.

---

> > > ### Author Response · Authors · 2024-12-03
> > > **Gentle Reminder**
> > >
> > > Dear Reviewer JXF8,
> > >
> > > Thank you once again for your valuable time. This is a gentle reminder that only **2 hours** remain to provide any additional comments or feedback. We sincerely hope our responses have addressed your concerns. If you need any further clarification, please don't hesitate to let us know—we're happy to assist. Otherwise, we kindly ask you to consider updating your evaluation accordingly.
> > >
> > > Your time and effort in reviewing our paper are deeply appreciated.

---

> ### Author Response · Authors · 2024-12-03
> **Appreciation and Final Remarks**
>
> Dear Reviewer JXF8,
>
> We are grateful for your invaluable time and thoughtful review.
>
> Throughout the discussion period, we clarified the experimental results on the base models to address your concern regarding pre-training data (Weakness 1) and provided further explanation on the core contribution of our study to address your concern regarding RT (Weakness 2). For your questions, we conducted additional experiments with the OLMo model, confirming results consistent with those in our paper (Question 1), and corrected the numerical errors you identified (Question 2).
>
> As the author response period comes to a close, we would like to inquire whether our responses have sufficiently addressed your concerns. If so, we kindly request that you consider updating your evaluation accordingly. While the reviewer response period has ended, we would like to remind you that a post-rebuttal review edit is still possible.
>
> We sincerely hope our efforts have addressed your concerns. Thank you once again for your dedication to reviewing our paper.

---

### Official Review · Reviewer_6b1i · 2024-11-04

**Soundness:** 3
**Presentation:** 3
**Contribution:** 3
**Rating:** 8
**Confidence:** 4

**Summary:**

This paper demonstrates that only fine-tuning base models on the responses in instruction datasets (response tuning, or RT) often results in comparable performance to fine-tuning on the responses *conditioned on the instructions* (traditional instruction tuning, or IT) without losing any core model capabilities. This supports the authors' hypothesis that models have already learned instruction following ability during pretraining, and instruction tuning thus mainly facilitates the model placing higher probabilities on tokens associated with helpful responses at inference time, or what the authors call "aligning the training response distribution". The authors measure response quality using 4 metrics: Likert-style individual response acceptability ratings + pairwise response preference selection using both human annotators and GPT4-as-a-judge. They find results to be consistent across 3 models from different families and on 3 instruction-tuning datasets, and also find that a larger Gemma model closes the gap between RT and IT more than a smaller one.

The authors further investigate whether using an LLM to refine the response quality along dimensions such as clarity, structure, and tone before training improves model response quality (as judged by GPT4), and find a positive result.

Finally, they extend their experimental setting to refusal, and show that only training on responses from refusal/safety datasets (and not the unsafe queries the responses are associated with) also yields comparable performance, without causing the model to over-refuse. This is particularly interesting, because it supports the hypothesis that models learn to identify unsafe queries during pretraining, but need tuning to learn the "language" of refusal.

**Strengths:**

- Originality: the paper takes a creative approach to an underexplored research question.
- Quality: The paper includes quite extensive experiments to answer 3 main RQs and is not lacking on content. The authors include multiple datasets and models in their analysis. I appreciate that they use both human and model evaluation that is generally well-controlled for biases (e.g., not using the same model/family to evaluate another model); given that both evaluation methods have known shortcomings, their results seem more robust due to the extensive effort given to evaluation.
- Clarity: The paper is generally well-written. Given there are a large amount of experimental results, the authors make good use of the Appendix for additional details while also making sure the main paper is readable and self-contained. I appreciate the delineation of the paper into 3 main sections corresponding to the 3 RQs of interest. Generally comprehensive lit review.
- Significance: The paper provides a useful datapoint an interesting insight to ongoing discussion about what models learn during instruction tuning. While the 2nd RQ is a bit orthogonal to the paper's narrative, the 1st and 3rd RQs are interesting and impactful. I would like to see it presented at the conference.

**Weaknesses:**

Main:
- My main qualm with the paper is the authors use the word "comparable" liberally when describing the performance of RT and IT tuning. However, in most cases, it is consistently the case that IT outperforms RT, often by what looks like rather substantial margins (e.g., Figures 2 and 4). I agree with the authors that it's really interesting that RT can make up the majority of the gap between the untuned and IT models, but I think the language can be toned down a bit and more discussion given to what performance aspects cannot be solely learned from RT alone. Alternatively, the authors could provide a quantitative definition of what they consider comparable performance.
Similarly, some of the differences in core capabilities (Table 1) seem meaningful (such as +7 on HellaSwag for Gemma 2 and -7 on GSM8k for Llama 3.1), but these are mostly chalked up to being typical or expected noise. It would be valuable to include a line for the untuned/base model in this table as a comparison point.
- The URIAL paper should certainly be discussed in the related work section as it asks a similar RQ and has similar conclusions to this paper (though different methods).
- The authors make a key assumption about the data they instruction- and response-tune on: that it hasn't been included in the pretraining data of the models in a meaningful way. This assumption seems to be supported by the poor performance of the untuned models in, e.g., Figure 2, but it should be stated more concretely.
- Practical impact: from an analysis/model understanding perspective, the paper is strong. The authors do not explain, however, what impacts only training on responses could have from a practical angle. Presumably, it is not cheaper to collect the response data, since human or synthetic data generation pipelines would still require the instructions or refusal query to be present. It could be faster to train the model when one does not have to run inference on the instruction tokens, though the loss computation is the same so this may be minor. I would appreciate some discussion about this, and/or quantitative #s on, e.g., finetuning speed, in the paper.

More minor:
- the authors use a number of terms in their main claims that are a bit vague and deserve proper definition, such as "output/response space", "training response distribution", "behavioral guide" and "inherent attributes" of the response space. I got the gist of what they were saying, but I don't think that should be assumed, and it would be good to be more consistent w/ terminology and more clearly define it.
- the paper could benefit if the authors added some analysis on models with open training data (such as Olmo or Pythia)-- this would allow them to perform a leakage analysis and understand what types of/how much instruction data is present in the pretraining corpus.
- Comparing Figs 3 and 6 it appears that both URIAL and URIAL-R are competitive with RT/IT, which raises the question of **why are we instruction-tuning in the first place?** I would appreciate some clarification of this in the rebuttal.

**Questions:**

Notes and minor other points:
- How are you testing the models' core capabilities in Table 1-- prompt format, instructions in prompt, greedy decoding, few-shot examples?
- typos line 044: "omits"; line 054 missing "we"; line 082: "guides"; line 092: "focuses"; line 405: missing "the"; line 406: extra "the"
- it would be good to link to the plots for the other models in the Appendix in the caption of each Figure, for reader's easy reference and cross-model comparison.
- related work: would be good to cite https://aclanthology.org/2020.emnlp-main.105/ for cross-task generalization as this was the earliest (?) work
- The authors should mention this concurrent work in a camera-ready: https://arxiv.org/abs/2409.14254
- I didn't find Figs 3/6 to be particularly useful and I am not sure what the 1-5 metric represented is. At a minimum it would be good to make the graphs represent the same metrics so that Figs 3 and 6 can be directly compared.
- Using the word "significant" to describe results (lines 312, 403) implies you've done statistical testing

---

> ### Author Response · Authors · 2024-11-22
> **Response to Reviewer 6b1i (1/2)**
>
> We sincerely appreciate your thoughtful and constructive feedback, as well as your recognition of the value of our work. We have addressed each of your comments below and incorporated the suggested revisions into the manuscript.
>
> ---
>
> **Response to main weaknesses:**
>
> **1. I agree with the authors that it's really interesting that RT can make up the majority of the gap between the untuned and IT models, but I think the language can be toned down a bit and more discussion given to what performance aspects cannot be solely learned from RT alone. Similarly, some of the differences in core capabilities seem meaningful. It would be valuable to include a line for the untuned/base model in this table as a comparison point.**
>
> Thank you for your insightful feedback. In response, we conducted a qualitative analysis of cases where the IT model outputs were preferred over RT outputs. The results are detailed in our general response 3 and have been incorporated into the updated manuscript (line 311).
>
> Additionally, we have included evaluation results for the untuned base models in Tables 1 and 11 to provide a clearer comparison point. We expanded our explanation of these results, noting that both IT and RT models exhibit a similar, slight performance drop (commonly referred to as the alignment tax [1]) while largely retaining the core capabilities of their pre-trained base models. However, upon reviewing Table 11, we observed that consistent trends across different models and datasets are limited.
>
> **2. The URIAL paper should certainly be discussed in the related work section.**
>
> Thank you for highlighting this relevant work. We have cited and mentioned the URIAL paper in the related work section of the revised manuscript (line 116).
>
> **3. The assumption that IT/RT data used hasn't been included in the pre-training data of the models in a meaningful way seems to be supported by the poor performance of the untuned models in, e.g., Figure 2, but it should be stated more concretely.**
>
> Thank you for highlighting this critical aspect of our experiments. In the revised manuscript, we have explicitly detailed our explanation of the results to address this assumption more concretely (line 303). We believe your suggestion will greatly assist readers in better interpreting the findings of our experiments.
>
> **4. From an analysis/model understanding perspective, the paper is strong. I would appreciate some practical aspects of RT, and/or quantitative #s on, e.g., finetuning speed, in the paper.**
>
> We are pleased that you value the contributions of our paper and appreciate your insightful suggestions. In response, we have addressed the points you raised, along with related suggestions from other reviewers regarding the practicality of RT, in the general response 2.

---

> > ### Author Response · Authors · 2024-11-22
> > **Response to Reviewer 6b1i (2/2)**
> >
> > **Response to minor weaknesses:**
> >
> > **1. It would be good to be more consistent w/ terminology and more clearly define it.**
> >
> > Thank you for pointing this out. We have standardized the term "output/response space" to the standard term "response distribution." We have revised "behavioral guide" to "behavioral guidance" and clarified "inherent attributes” (line 363).
> >
> > **2. the paper could benefit if the authors added some analysis on models with open training data (such as Olmo or Pythia)-- this would allow them to perform a leakage analysis and understand what types of/how much instruction data is present in the pretraining corpus.**
> >
> > Thank you for the thoughtful suggestion. In response, we fine-tuned and evaluated OLMo trained with LIMA, observing that the trend reported in the paper remains consistent for this open-data model. The results are provided in our general response 4.
> >
> > Our intuition motivating this study is that language models may implicitly learn instruction-following dynamics from texts in community threads, such as StackOverflow, and may develop the capabilities to identify unsafe requests from sources like news or legal articles in their pre-training corpora. Using [2], we identified that popular open pre-training corpora frequently contain such data. We have incorporated this evidence into the revised manuscript to further support and clarify the motivation behind our study (line 037).
> >
> > **3. Comparing Figs 3 and 6 it appears that both URIAL and URIAL-R are competitive with RT/IT, which raises the question of why are we instruction-tuning in the first place? I would appreciate some clarification of this in the rebuttal.**
> >
> > Thank you for the question. While URIAL has demonstrated promising results in recent studies, IT remains the widely adopted standard for alignment due to its established effectiveness and broad acceptance in practical applications. URIAL, on the other hand, is still in an exploratory phase, and its potential and limitations are actively being investigated [3]. We view URIAL as a promising area for further research and included it in our design for the comprehensive study and to encourage future exploration.
> >
> > **Response to questions and suggestions:**
> >
> > **1. How are you testing the models' core capabilities in Table 1-- prompt format, instructions in prompt, greedy decoding, few-shot examples?**
> >
> > Thank you for requesting clarification. For the evaluations, we used open-source packages—open-Instruct and lm-eval—with greedy decoding. For zero-shot evaluations, each instance is formatted as a user message, and the answer options are inserted as assistant responses. The option with the highest log probability is considered as the model's choice. For the GSM8K few-shot evaluation, we follow [4], using an 8-turn (i.e. shot) chain-of-thought prompt as input, with the predicted label extracted using regular expressions. We have updated the manuscript to link the core capabilities evaluation setup in the main text to the detailed settings in Appendix B (line 296).
> >
> > **2. Addressing writing suggestions**
> >
> > We sincerely appreciate your detailed suggestions. Following your advice, we have corrected the identified typos and added links to the full results in the captions of the result figures and tables in the main content to facilitate easier cross-referencing (line 231, 248, 281, 284). Additionally, we have cited the relevant works you referenced (line 108, 116) and plan to include a discussion of concurrent work in the camera-ready version. Regarding the JustEval metrics and figures, we have clarified the evaluation criteria: scores from 1 to 5 represent the level of agreement of an LLM judge on how well the model's response satisfies the metric [5] (line 246). The difference between Figures 3 and 6 lies in the inclusion of the safety metric. Since the experiments in Section 4 did not involve safety tuning, evaluations on these metrics were not included (whereas URIAL and URIAL-R include safety-focused examples). Finally, we replaced 'significant' with 'substantial' to enhance precision.
> >
> > ---
> >
> > **References**
> >
> > [1] Fu et al. "Disperse-Then-Merge: Pushing the Limits of Instruction Tuning via Alignment Tax Reduction." Findings of ACL 2024.
> >
> > [2] Elazar et al. "What's In My Big Data?." ICLR 2024. (https://wimbd.apps.allenai.org/)
> >
> > [3] Zhao et al. "Is In-Context Learning Sufficient for Instruction Following in LLMs?" arXiv preprint. 2024.
> >
> > [4] Dubey et al. "The llama 3 herd of models." arXiv preprint. 2024.
> >
> > [5] Lin et al. "The unlocking spell on base llms: Rethinking alignment via in-context learning." ICLR 2024.

---

> ### Author Response · Authors · 2024-11-28
> **Looking Forward to Further Discussions**
>
> Dear Reviewer 6b1i,
>
> We sincerely appreciate the time and effort you have devoted to reviewing our paper and providing valuable feedback. We have carefully considered your comments and endeavored to address your concerns and questions. We kindly ask if our responses and the updated manuscript adequately address them. If so, we would like to respectfully request your reconsideration of the rating.
>
> If there are any areas requiring further clarification or additional explanation, please do not hesitate to let us know. We would be happy to address any additional comments you may have. Thank you once again for your thoughtful review and efforts.

---

> > ### Comment · Reviewer_6b1i · 2024-11-28
> >
> > Hello,
> >
> > Thank you for the detailed response and updated paper draft. I particularly appreciate the efforts you have put in to add qualitative examples and demonstrate the practicality of RT. I will maintain my positive score.

---

> ### Author Response · Authors · 2024-11-29
> **Appreciation and Final Remarks**
>
> Thank you for reviewing our responses and updated paper. We truly appreciate your positive assessment and are grateful for your thoughtful support.

---

### Official Review · Reviewer_AfZU · 2024-11-04

**Soundness:** 3
**Presentation:** 4
**Contribution:** 3
**Rating:** 5
**Confidence:** 5

**Summary:**

The main question this paper tries to answer is "how much do we lose by tuning just on the responses (response tuning) instead of instruction-response pairs (instruction tuning)". The authors use existing IT datasets and remove instructions from them to get datasets for response tuning. They fine-tune a variety of base-llms (not instruction tuned) and compare IT vs RT. They evaluate instructability with human evaluation, GPT-4 based response quality evaluation and pairwise assessment. They also evaluate core capabilities across standard benchmarks like MMLU, OpenbookQA, HellaSwag, ARC, GSM 8K, PIQA. Across a wide range of experiments, they show that response tuning comes close to instruction tuning. Additionally, they show the application of this method for preference alignment and safety alignment. Finally, they show that a similar trend holds for in-context learning, where using just responses doesn't degrade performance significantly compared to using instruction-response pairs.

**Strengths:**

* originality - The authors ask a simple question that hasn't been considered from a scientific point of view. "Can we use good responses instead of instruction-response pairs"
* quality - The authors run a wide variety of experiments to make their claim stronger and well substantiated.
* clarity - The paper is well-written and easy to read. In most places all the necessary details and citations are provided
* significance - There is the scientific question on the source of gains due to instruction tuning. From a practical perspective, this raises the question whether, as a community, we should invest more energy looking at instruction-free response datasets which are free of instruction biases.

**Weaknesses:**

There are two main concerns regarding the soundness of these experiments
## Leaking instructions to response
The datasets in section 4, 5 and 6 are collected as instruction - response pairs. The data generation process may induce a leak from the instruction to the response. For e.g. if the instruction was "Write an article on the role of dinosaurs in prehistoric ecology", and the response was "Dinosaurs played a significant role in prehistoric ecology ...", the instruction can be inferred based on the response. In a trivial situation, for an instruction "Follow the rules x, y, z", the response could be "following the rules x, y, z, here's the response ...". To argue that one needs only the response (and not the instruction), the authors would need to show that the instruction cannot be trivially inferred from the response. Alternatively, an experiment training on high quality responses gathered without explicit instructions could also prove the point.

## Training setup (Section 4.1)
The authors use PEFT (QLoRA) but don't give any references that this is a standard setup for instruction tuning. It is possible that a less expressive training scheme (PEFT vs full fine-tuning) suppresses the utility of having instructions in the training data. Alternatively, if they could show that their PEFT-trained IT models are comparable to the best IT models on these datasets, that could alleviate this concern too.

**Questions:**

My two main questions with potential fixes are indicated in the weaknesses section. Convincing experimental results for the two weaknesses will change my stance on soundness and the overall rating.

Some suggestions that aren't critical to soundness
* "establishing an output space" is less standard than "establishing an output distribution". The authors do use "distribution" at various places in the paper, so it isn't clear why not use the same standard term everywhere (including the abstract)
* The authors use certain terms that aren't well-defined. 073 - "contextual refusals",
* line 311 - "some fluctuations" - why are the fluctuations attributed to fine-tuning? if there's a good reason, it should be stated, if not, it is sufficient to state that the numbers are comparable.
* line 333 - what does the up arrow mean?
* line 428 - what exactly is meant by "pattern matching"?
- Figure 4b - IT win-rate is 60% compared to RT win rate of 40%. This is a big difference but there is no discussion of this negative result in the text.

The authors consistently use the narrative that "capabilities required to behave as a chat assistant are largely inherent in pre-trained LLMs and can be activated without explicit supervision from instruction-response pairs" (lines 096, 515 and perhaps more). However, there is no direct proof for the "capabilities being inherent in pre-trained LLMs". One can argue that training on responses adds these capabilities that aren't inherent, just like instruction-response pairs are supposedly adding these capabilities. This claim is unnecessary and unsubstantiated. This claim is unnecessarily distracting and I would urge the authors to stay focused on the central claim of response tuning being at-par with instruction tuning.

---

> ### Author Response · Authors · 2024-11-22
> **Response to Reviewer AfZU (1/2)**
>
> We deeply appreciate your thoughtful feedback. Your comments have been invaluable in enhancing the rigor and clarity of our work. Below, we address each concern and suggestion, with corresponding updates reflected in the revised manuscript.
>
> ---
>
> **Response to weaknesses:**
>
> **1. Potential instruction leakage into responses**
>
> Thank you for raising this valid concern. To verify whether RT is effective with data free from instruction leakage, we trained the model using 1,024 pure Python code snippets as response data. These snippets were extracted from the StarCoder IT dataset [1], but we only used code devoid of comments or docstrings to ensure their purity. This cleaned dataset is intrinsically leakage-free, as one cannot easily figure out the original query from code snippets.
>
> We then evaluated the RT model’s ability to follow code generation instructions using the HumanEvalPack benchmark [2]. The results, shown below, indicate that even when trained with data free from instruction leakage, the RT model successfully follows code generation instructions:
>
> |  | IT  | RT  |
> | --- | --- | --- |
> | Pass@10 | 0.45 | 0.40 |
>
> We would like to note that this experiment was originally suggested by Reviewer krvS to demonstrate the practicality of RT. However, we extended this scenario to a pure-code setting specifically to address your concern regarding instruction leakage.
>
> **2. Training setup**
>
> To address your concern about the training setup, we conducted full fine-tuning of the Llama-3.1-8B model on the LIMA dataset and evaluated both IT and RT models. The results indicate that PEFT models achieve comparable performance to full fine-tuning, and the trend of pairwise evaluations between IT and RT models remains consistent:
>
> |  | JustEval overall score | Win-rate against IT counterparts |
> | --- | --- | --- |
> | IT (QLoRA) | 3.88 | - |
> | RT (QLORA) | 3.70 | 40.1% (vs QLoRA IT) |
> | IT (Full-FT) | 3.74 | - |
> | RT (Full-FT) | 3.49 | 40.5% (vs Full-FT IT) |
>
> Additionally, prior studies validate that QLoRA achieves performance comparable to 16-bit full-parameter fine-tuning across various metrics [3]. For instance, the instruction-following Guanaco model, trained with QLoRA on the Oasst1 IT dataset, was among the top-performing models of its time [3].

---

> ### Author Response · Authors · 2024-11-22
> **Response to Reviewer AfZU (2/2)**
>
> **Response to questions and suggestions:**
>
> **1. On the use of terms.**
>
> Thank you for the constructive feedback. In the revised manuscript, we have replaced "output/response space" with "response distribution" for consistency. Additionally, we have replaced "contextual refusals" with "explanatory refusals" and clarified the term (lines 077, 434). The up arrow indicates "higher is better," and "pattern matching" has been replaced with its meaning, "keyword searches" (line 437).
>
> **2. Why are the fluctuations attributed to fine-tuning? if there's a good reason, it should be stated, if not, it is sufficient to state that the numbers are comparable.**
>
> Thank you for the suggestion. In this context, “fluctuation” refers to the performance discrepancies observed after fine-tuning, also known as alignment tax [4]. In the revised manuscript, we have included evaluation results for the base model and expanded our explanation to provide greater clarity (line 303). Our findings indicate that both IT and RT models exhibit a similar and slight level of performance drop after fine-tuning, while largely retaining the core capabilities of their pre-trained base models.
>
> **3. IT win-rate is 60% compared to RT win rate of 40%. This is a big difference but there is no discussion of this negative result in the text.**
>
> Thank you for raising this point. We agree that further discussion of these results would add value to our work. In response, we conducted a qualitative analysis of the cases where the IT model was preferred over the RT model in pairwise evaluations. This analysis is detailed in our general response 3, and the findings have been incorporated into the revised manuscript (line 311).
>
> **4. There is no direct proof for the "capabilities being inherent in pre-trained LLMs". This claim is unnecessarily distracting and I would urge the authors to stay focused on the central claim of response tuning being at-par with instruction tuning.**
>
> We appreciate your valuable suggestion and would like to offer additional context regarding this narrative. While we acknowledge that the claim of “inherent capabilities” may not be directly provable, our hypothesis builds on prior studies suggesting that pre-training functions as massive multi-task learning [5]. Extending this view, we posit that models may implicitly learn instruction-following dynamics from texts in community threads, such as StackOverflow, and may develop the capabilities to identify unsafe requests from sources like news or legal articles in their pre-training corpora [6].
>
> Given the scale disparity between pre-training (15 trillion tokens) and post-training (573K tokens for LIMA responses), and the explicit exclusion of instructions during supervised fine-tuning, we believe our empirical results, including those addressing weakness 1, provide indirect support for this hypothesis. Thus, we put this explanation as a plausible reason why pre-trained LLMs require minimal fine-tuning to effectively transition into instruction-following and safe assistants. To clarify our rationale, we have incorporated this additional context into the revised manuscript (line 037).
>
> ---
>
> **References**
>
> [1] Wei et al. "SelfCodeAlign: Self-Alignment for Code Generation." NeurIPS 2024.
>
> [2] Muennighoff et al. "Octopack: Instruction tuning code large language models." ICLR 2024.
>
> [3] Dettmers et al. "Qlora: Efficient finetuning of quantized llms." NeurIPS 2023.
>
> [4] Fu et al. "Disperse-Then-Merge: Pushing the Limits of Instruction Tuning via Alignment Tax Reduction." Findings of ACL 2024.
>
> [5] Radford et al. "Language models are unsupervised multitask learners.”
>
> [6] Elazar et al. "What's In My Big Data?." ICLR 2024. (https://wimbd.apps.allenai.org/)

---

> ### Author Response · Authors · 2024-11-28
> **Looking Forward to Further Discussions**
>
> Dear Reviewer AfZU,
>
> We sincerely appreciate the time and effort you have devoted to reviewing our paper and providing valuable feedback. We have carefully considered your comments and endeavored to address your concerns and questions. We kindly ask if our responses and the updated manuscript adequately address them. If so, we would like to respectfully request your reconsideration of the rating.
>
> If there are any areas requiring further clarification or additional explanation, please do not hesitate to let us know. We would be happy to address any additional comments you may have. Thank you once again for your thoughtful review and efforts.

---

> > ### Comment · Reviewer_AfZU · 2024-11-30
> >
> > Thank you for the response and updates to the manuscript. I'm happy with the update regarding Training setup and other misc. items.
> >
> > > Potential instruction leakage into responses
> > I don't think this is sufficiently addressed and has the potential to invalidate the results of the paper as a whole. Imagine a model, finetuned to predict the instruction given the response, and look at how predictable the instruction is given the response for each of the evaluation datasets. My guess would be that the instructions are highly predictable from responses across various datasets. If that's the case, then effectively, training on the responses is not much different than training on instruction-response pairs, in terms of the data itself.
> >
> > The experiment with HumanEvalPack benchmark isn't sufficient either. Note that the responses, even when stripped of doc-strings and comments, are still responses to instructions in the form of code. They may still have function names, variable names, function calls that give the instruction away.
> >
> > Also the baseline performance of 0 with base models for HumanEvalPack benchmark (as mentioned in the response to another reviewer) feels suspiciously bad to me and I can't trust this particular result. For instance, if I ran llama3.1 8-bit quantized model on my laptop with ollama (`ollama run llama3.1:8b-text-q8_0`) and appended ` ```python` to the instruction, I get a reasonable looking python program and not meaningless output tokens. I feel certain that a silly fix like this is all that's needed to bet the base model to produce something more meaningful. (Note that in absense of ```python, all I got as a response was \n)
> >
> > ```
> > >>> Write a Python function `has_close_elements(numbers: List[float], threshold: float) -> bool` to solve the following problem:\nCheck if in given list of numbers, are any two numbers closer
> > ... to each other than given threshold.\n>>> has_close_elements([1.0, 2.0, 3.0], 0.5)\nFalse\n>>> has_close_elements([1.0, 2.8, 3.0, 4.0, 5.0, 2.0], 0.3)\nTrue. ```python
> >  \n def has_close_elements(numbers: List[float], threshold: float) -> bool:\n if not numbers or len(numbers) < 2:\n return False\n for i in range(len(numbers)-1):\n for j in range(i+1,
> > len(numbers)):\n if abs(numbers[i] - numbers[j]) <= threshold:\n return True\n return False ```
> > ```
> >
> > As I mentioned in my review, I can think of only two ways to remove this confounder. Either, collect high quality responses, to model the response distribution. In this case it could be high quality code snippets completely independent of instructions (perhaps some public code from high quality packages from github). Or take instruction tuning datasets, train a model to predict probability of instruction given response and from the RT dataset, remove responses that are predictive of the instructions beyond a certain threshold.
> >
> > I will keep my rating unchanged.

---

> > > ### Author Response · Authors · 2024-12-02
> > > **Follow-Up Response to Reviewer AfZU (1/2)**
> > >
> > > Thank you for your valuable comment and engagement. We would like to address your concerns below.
> > >
> > > ---
> > >
> > > **1. On the concern of instruction leakage**
> > >
> > > Thank you for your clarification. We now understand the potential leakage concern you raised. In light of this, we would like to provide additional experimental results.
> > >
> > > To address your concern, we randomly excerpted 1,000 passages from news articles (CC-News [1]) and used them as RT training data after processing. News articles are not generated from instructions, so they inherently lack instructions that could be leaked. Furthermore, inferring a hypothetical instruction from a snippet of a news article might not be trivial (please see the examples below). To establish an adequate response distribution using news articles, we cleaned the excerpted sentences, enhanced their readability, and adjusted their tone using an LLM (GPT-4o). The prompt and examples of the training data can be found below.
> > >
> > > Using this dataset, we fine-tuned Llama-3.1-8B and assessed its instructability using JustEval. Since it does not have a pair-tuned counterpart, we include the performance of our Llama-3.1-8B LIMA IT/RT models as a point of comparison. The results are as follows:
> > >
> > > |  | JustEval overall score |
> > > | --- | --- |
> > > | Llama-3.1-8B News RT | 3.35 |
> > > | Llama-3.1-8B LIMA RT | 3.70 |
> > > | Llama-3.1-8B LIMA IT | 3.88 |
> > > | Llama-3.1-8B Base | 2.16 |
> > >
> > > The results demonstrate that the RT model trained using segments of news articles, while its performance is somewhat inferior to the model trained on diverse responses, can still function as an instruction-following model, supporting our claims. Examples of the model's outputs can be found below. We hope these results address your concerns.
> > >
> > > ---
> > >
> > > ```
> > > # Extraction and refinement prompt
> > >
> > > Your task is to process a raw news article in two steps: Extraction and Refinement.
> > >
> > > 1. Extraction: Randomly select a portion of the news article. This can include one or more paragraphs or a set of sentences.
> > > 2. Refinement: Edit the extracted text to enhance readability and presentation:
> > >     - Remove any extraneous elements, such as headings, symbols, disclaimers, or other non-content components.
> > >     - Reformat the text for better readability. You may use structured formats if they enhance readability.
> > >     - Adjust the tone to a friendly and conversational assistant style.
> > >
> > > Steps for Processing:
> > > - Randomly select a portion of the news article and write it first.
> > > - Refine the extracted text as described above. Present your refined response in this format: "Refined news: [Your improved version of the text]."
> > >
> > > Do not include any additional explanations or notes after "Refined news:".
> > > Now, process the following news article:
> > > <BEGIN NEWS>{news}<END NEWS>
> > > ```
> > >
> > > ```
> > > # Training data example 1
> > >
> > > Two-thirds of businesses expressed enthusiasm about employees sharing job vacancies and company news on social media. However, this approach is seen as a limited form of social advocacy. The real impact comes from employees sharing authentic, personal insights about their workplace. Despite this, many companies hesitate to allow such unfiltered content, citing concerns over brand control. Over half of the surveyed businesses admitted they lack the time to monitor employee posts. The study also revealed that 35% of companies permit employees to post on social media but keep a close watch on their activity, while 32% either restrict or tightly regulate such posts.
> > > ```
> > >
> > > ```
> > > # Training data example 2
> > >
> > > A small sculpture of a girl has captured attention at the Scott Monument in Edinburgh. Inspired by Banksy’s iconic “Girl With The Balloon,” this new artwork, titled "Mine Girl," offers a unique twist. The girl is depicted standing near the monument’s plinth, gazing up at a bright red, mine-shaped balloon with a heart cut out. Instead of a traditional string, a black chain connects the balloon to her hand, adding a striking and thought-provoking element to the piece.
> > > ```
> > >
> > > ```
> > > # Model output example 1
> > >
> > > Instruction
> > > ------------
> > > How do bees make honey?
> > >
> > > Model Response
> > > ------------
> > > Bees collect nectar from flowers, which contains a large amount of sugar. To store this sugar, bees use a special stomach organ called a honey stomach. When a bee returns to the hive, it regurgitates the nectar into the honeycomb cells, where other worker bees help to evaporate the excess water. This process gradually thickens the nectar into a rich, golden honey.
> > > ```
> > >
> > > ```
> > > # Model output example 2
> > >
> > > Instruction
> > > ------------
> > > How do you make a classic French omelette?
> > >
> > > Model Response
> > > ------------
> > > First, heat some butter in a non-stick pan over medium-high heat. Once the butter has melted and is sizzling, carefully pour in the egg mixture. Immediately start drawing the edges of the eggs toward the center, tilting the pan to allow the uncooked egg to flow in. Continue this motion until the eggs are set but still glossy and creamy in the center.
> > > ```

---

> > > > ### Author Response · Authors · 2024-12-02
> > > > **Follow-Up Response to Reviewer AfZU (2/2)**
> > > >
> > > > **2. On base model performance on the HumanEvalPack benchmark**
> > > >
> > > > Thank you for raising this concern. Since all test instructions specify “`Write a Python function`,” providing a generation prefix that forces the model to generate the expected output (in this case, Python code), such as ` ```python ` (Markdown Python code block opening quotes), can be interpreted as giving a partial response. We believe this might be considered unfair when evaluating the instructability of the models. Therefore, we did not provide such a prefix for any of the models and reported the evaluation results based on their raw outputs. In this setup, we did not observe any valid or correct code from the base model.
> > > >
> > > > ---
> > > >
> > > > **References**
> > > >
> > > > [1] https://huggingface.co/datasets/vblagoje/cc_news

---

> > > > > ### Author Response · Authors · 2024-12-02
> > > > > **Awaiting Your Valuable Feedback**
> > > > >
> > > > > Dear Reviewer AfZU,
> > > > >
> > > > > We deeply appreciate the time and effort you’ve dedicated to the review process. This is a gentle reminder that there are 24 hours remaining for you to provide any additional comments or feedback on our paper.
> > > > >
> > > > > We look forward to receiving your further comments. If there are any remaining questions or areas requiring clarification, please don’t hesitate to share them—we would be happy to provide any additional information you may need. If our previous responses have sufficiently addressed your concerns, we kindly request you to consider updating your evaluation.
> > > > >
> > > > > Thank you once again for your thoughtful contributions.

---

> > > > > > ### Author Response · Authors · 2024-12-03
> > > > > > **Gentle Reminder**
> > > > > >
> > > > > > Dear Reviewer AfZU,
> > > > > >
> > > > > > Thank you once again for your valuable time. This is a gentle reminder that only **2 hours** remain to provide any additional comments or feedback. We sincerely hope our responses have addressed your concerns. If you need any further clarification, please don't hesitate to let us know—we're happy to assist. Otherwise, we kindly ask you to consider updating your evaluation accordingly.
> > > > > >
> > > > > > Your time and effort in reviewing our paper are deeply appreciated.

---

> > > > > > > ### Author Response · Authors · 2024-12-03
> > > > > > > **Appreciation and Final Remarks**
> > > > > > >
> > > > > > > Dear Reviewer AfZU,
> > > > > > >
> > > > > > > Thank you very much for your engagement and thoughtful feedback throughout the review process. We are grateful for your recognition of the value of our submission and your acknowledgment of our initial responses and the updated manuscript.
> > > > > > >
> > > > > > > As the author response period comes to a close, we would like to ask if our additional experimental results on RT using the news dataset address your remaining concerns regarding instruction leakage. If so, we kindly request that you consider updating your evaluation accordingly. While the reviewer response period has ended, we would like to remind you that a post-rebuttal review edit is still possible.
> > > > > > >
> > > > > > > We sincerely hope that our efforts have addressed your concerns and look forward to any further feedback you may have. Thank you once again for your invaluable time and dedication.

---

### Author Response · Authors · 2024-11-22
**General Response to All Reviewers (1/2)**

We sincerely appreciate the reviewers for their constructive comments and valuable suggestions, which have greatly helped us improve our manuscript. Below, we address common concerns and questions raised and summarize the revisions implemented in response.

**1. Scope and goal of our study**

We would like to clarify the scope and goal of our study. The primary aim of our work is to contribute insights into the mechanisms by which language models develop the capabilities needed to handle instructions helpfully and safely. RT serves as a tool in our study, intentionally designed as a simple ablation method of IT. As highlighted by Reviewer 6b1i and krvS, the value of our work lies in the experimental findings and insights derived from them, rather than RT being proposed as a standalone method. We believe these findings are valuable and worth sharing.

**2. Practicality of RT**

While the primary focus of our study is to deepen the understanding, we recognize the importance of dicussing the practical aspects of RT to stimulate future exploration.

**2.1. RT for domains where instruction-response pairs are hard to obtain**

Reviewer krvS raised an important question about whether RT could be effective in domains where instruction-response pairs are challenging to gather, yet specific response formats are necessary—such as in code generation. To investigate this, we fine-tuned the Llama-3.1-8B model using 1,024 pure Python code snippets as RT training data. These snippets were sourced from the StarCoder IT dataset [1], but we only used code devoid of comments or docstrings to simulate data that is easier to collect. We then evaluated the model’s ability to follow code generation instructions using the HumanEvalPack benchmark [2]. The results demonstrate that the RT model, trained solely on pure code, performs comparably to the IT model in adhering to code generation instructions.

|  | IT  | RT  |
| --- | --- | --- |
| Pass@10 | 0.45 | 0.40 |

**2.2. Reduced computational cost**

In addition, as noted by reviewer 6b1i, RT generally requires fewer tokens, which results in a reduced number of floating-point operations compared to IT, thereby leading to shorter training times. For instance, the number of floating-point operations performed during fine-tuning Llama-3.1-8B is shown below:

|  | IT | RT |
| --- | --- | --- |
| Alpaca | 7.39e+18 | 6.87e+18 (-8%) |
| Dolly | 3.73e+18 | 1.31e+18 (-65%) |
| LIMA | 6.78e+17 | 5.09e+17 (-25%) |

These findings have been incorporated into the revised manuscript (Appendix A and E).

**3. Additional analysis of pairwise evaluations**

Reviewers 6b1i, JXF8, and krvS suggested further discussion of the pairwise evaluation results. We agree that analyzing these results adds depth and value to our work. In response, we conducted a qualitative analysis of cases where outputs from IT models are preferred over those of RT models finds that IT models often produce outputs with a style more closely aligned with the given instructions. This suggests that the explicit instruction-response mapping aids the model in selecting a response format more closely tailored to the instruction from the established response distribution. These have been incorporated into the revised manuscript (line 303), and we have adjusted the tone of our discussion to provide a more balanced perspective, as recommended by Reviewer 6b1i.

**4. Results for the open-data models**

To address suggestions from reviewers 6b1i, JXF8, and krvS, we also fine-tuned the OlMo 7B model using the LIMA dataset, along with 500 safety examples used only for safety evaluation. The results for this open-data model are consistent with the trends observed in our paper.

|  | AlpacaEval (vs IT win-rate) | AdvBench Refusal Rate |
| --- | --- | --- |
|  OLMo IT | - | 0.97 |
| OLMo RT | 37 % | 0.68 |

---

> ### Author Response · Authors · 2024-11-22
> **General Response to All Reviewers (2/2)**
>
> **5. Implemented Revisions**
>
> The following improvements have been incorporated into the manuscript, with changes highlighted in olive where applicable:
>
> - Corrected identified typos.
> - Corrected errors in average calculations for Tables 1 and 11 (previously omitted OpenBookQA).
> - **For Reviewers AfZU, 6b1i:** Standardized terminology and clarified definitions (e.g., "output space" → "response distribution”).
> - **For Reviewers AfZU, 6b1i, JXF8, krvS:** Conducted a qualitative analysis of pairwise evaluation results and incorporated the findings into the revised manuscript (Section 4, line 311).
> - **For Reviewers AfZU, 6b1i, krvS:** Added core capability evaluation results for the base models (Tables 1 and 11).
> - **For Reviewers AfZU, JXF8, krvS:** Expanded explanations of the motivation, scope, and goals of our study (Section 1 (line 037), Section 8 (line 537) and Appendix A).
> - **For Reviewer 6b1i:** Added relevant works to the Related Work section (Section 2, line 115), linked full evaluation results to captions in the main content (Section 4, line 231, 249, 281, 284), and included a description of the JustEval metric in the caption (Section 4, line 246).
> - **For Reviewers 6b1i, JXF8:** Expanded explanations of instructability evaluation results for the untuned models (Section 4, line 303).
> - **For Reviewers 6b1i, JXF8, krvS:** Added a discussion on practicality of RT (Appendix A and E).
>
> ---
>
> **References**
>
> [1] Wei et al. "SelfCodeAlign: Self-Alignment for Code Generation." NeurIPS 2024.
>
> [2] Muennighoff et al. "Octopack: Instruction tuning code large language models." ICLR 2024.

---

### Meta-Review · Area_Chair_9VQH · 2024-12-19

**Metareview:**

This paper proposes and studies response tuning, where an LM is aligned to become an instruction following model via just being finetuned on responses. The paper finds that in many cases, tuning on responses only can result in performant instruction following models.

On the positive side the paper presents an interesting approach that sometimes works well. On the negative side, there were some concerns with regard to the practicality of the method, the underperformance of the method vs. vanilla instruction tuning, as well as the experimental setup/datasets. In my view I do not think the results are interesting/significant enough to overcome the impracticality of the method, so I am recommending that this paper not be accepted.

**Additional Comments On Reviewer Discussion:**

Reviewer AfZU raised an important point about potential label leakage, which was addressed to an extent by the authors (but not convincingly enough, in my opinion). Several reviewers asked for experiments on more models, and this point was adequately addressed during the rebuttal. But in my opinion this was not enough to overcome the questions regarding the (imp)practicality of the method, which was raised by several reviewers (6b1i, krvS).

---

### Decision · Program_Chairs · 2025-01-22

Reject